# An analytic approximation of the feasible space of metabolic networks

Alfredo Braunstein[1,2,3], Anna Paola Muntoni[1] & Andrea Pagnani[1,2,4]

Assuming a steady-state condition within a cell, metabolic fluxes satisfy an underdetermined linear system of stoichiometric equations. Characterizing the space of fluxes that satisfy such equations along with given bounds (and possibly additional relevant constraints) is considered of utmost importance for the understanding of cellular metabolism. Extreme values for each individual flux can be computed with linear programming (as flux balance analysis), and their marginal distributions can be approximately computed with Monte Carlo sampling. Here we present an approximate analytic method for the latter task based on expectation propagation equations that does not involve sampling and can achieve much better predictions than other existing analytic methods. The method is iterative, and its computation time is dominated by one matrix inversion per iteration. With respect to sampling, we show through extensive simulation that it has some advantages including computation time, and the ability to efficiently fix empirically estimated distributions of fluxes.

[1] DISAT, Politecnico di Torino, 10129 Torino, Italy. [2] Human Genetics Foundation–Torino, 10126 Torino, Italy. [3] Collegio Carlo Alberto, 10024 Moncalieri, Italy. [4] Istituto Nazionale di Fisica Nucleare (INFN) Via Pietro Giuria, 110125, Torino, Italy. Correspondence and requests for materials should be addressed to A.B. (email: alfredo.braunstein@polito.it).

The metabolism of a cell entails a complex network of chemical reactions performed by thousands of enzymes that continuously process intake nutrients to allow for growth, replication, defence and other cellular tasks[1]. Thanks to the new high-throughput techniques and comprehensive databases of chemical reactions, large-scale reconstructions of organism-wide metabolic networks are nowadays available. Such reconstructions are believed to be accurate from a topological and stoichiometric viewpoint (for example, the set of metabolites targeted by each enzyme, and their stoichiometric ratio). For the determination of reaction rates, large-scale constraint-based approaches have been proposed[2]. Typically, such methods assume a steady-state regime in the system where metabolite concentrations remain constant over time (mass-balance condition). A second type of constraints limit the reaction velocities and their direction. In full generality, the topology of a metabolic network is described in terms of the chemical relations between the $M$ metabolites and $N$ reactions. In mathematical terms, we can define a $M \times N$ stoichiometric matrix $S$ in which rows correspond to the stoichiometric coefficients of the corresponding metabolites in all reactions. A positive (resp. negative) $S_{ij}$ term indicates that metabolite $i$ is created (resp. consumed) by reaction $j$. Assuming mass-balance and limited interval of variation for the different reactions, we can cast the problem in terms of finding the set of fluxes $\boldsymbol{v} \in \mathbb{R}^N$ compatible with the following linear system of constraints and inequalities:

$$S\boldsymbol{v}=\mathbf{b} \tag{1}$$

$$\boldsymbol{v}^{\mathrm{inf}} \leq \boldsymbol{v} \leq \boldsymbol{v}^{\mathrm{sup}} \tag{2}$$

where $\mathbf{b} \in \mathbb{R}^M$ is the known set of intakes/uptakes, and the pair $\boldsymbol{v}^{\mathrm{inf}}$, $\boldsymbol{v}^{\mathrm{sup}}$ represent the extremes of variability for the variables of our problem. Only in few cases, the extremes are experimentally accessible, in the remaining ones they are fixed to arbitrarily large values. It turns out that $N \geq M$, and the system is typically underdetermined. As an example, the RECON1 model of Homo sapiens has $N = 2,469$ fluxes (that is, variables) and $M = 1,587$ metabolites (that is, equations). The mass-balance constraints and the flux inequalities encoded in equations (1 and 2) define a convex-bounded polytope, which constitutes the space of all feasible solutions of our metabolic system.

The most widely used technique to analyse fluxes in large-scale metabolic reconstruction is flux balance analysis (FBA)[3,4] where a linear objective function, typically the biomass or some biological proxy of it is introduced, and the problem reduces to find the subspace of the polytope, which optimizes the objective function. If this subspace consists in only one point, the problem can be efficiently solved using linear programming. FBA has been successfully applied in many metabolic models to predict specific phenotypes under specific growth condition (for example, bacteria in the exponential growth phase). However, if one is interested in describing more general growth conditions, or is interested in other fluxes than the biomass[5], different computational strategies must be envisaged[6–8].

As long as no prior knowledge is considered, each point of the polytope is an equally viable metabolic phenotype of the biological system under investigation. Therefore, being able to sample high-dimensional polytopes becomes a theoretical problem with concrete practical applications. From a theoretical standpoint, the problem is known to be #P-hard[9] and thus an approximate solution to the problem must be sought. A first class of Monte Carlo–Markov chain sampling techniques available to analyse large-dimensional polytopes was originally proposed three decades ago[10] and falls under the name of Hit-and-Run

(HR)[11]. Basically, it consists on iteratively collecting samples by choosing random directions from a starting point belonging to the polytope. Unfortunately, polytopes defined by large-scale metabolic reconstructions are typically ill conditioned (that is, some direction of the space are far more elongated than others), and improved HR techniques to overcome this problem have been proposed[12] and implemented in the context of metabolic modelling[6,8,13]. Despite the fact that these dynamic sampling strategies are often referred as uniform random samplers, the uniformity of the sampling is guaranteed only in an asymptotic sense, and often establishing in practice how long a simulation should be run and how frequently the measurement should be taken for a given instance of the problem requires extensive preliminary simulations, which make their use very difficult under general conditions. Note, also that the problem of assessing how perturbations of network parameters affect the structure of the polytope is often of practical importance; for example, changing extremal flux values for studying growth rate curves or enzymopaties[14]. In these situations, in principle, the convergence time of the algorithm should be established independently for each new value of the parameter. Another limitation of this class of sampling strategies is the difficulty of imposing other constraints[15] such as the experimentally measured distribution profiles of specific subset of fluxes (typically biomass and/or in-take/out-take of the network), a particularly timely issue given the recent breakthrough of metabolic measurements in single cell[16], although recent attempts in this direction exist[17,18].

Recently, alternative statistical methods based on message passing (MP) techniques (also known as cavity or Bethe approximation in the context of statistical mechanics)[19] have been proposed[7,20–23], allowing for sampling of the polytope orders of magnitude faster than HR methods, under two main conditions: (i) the graphical structure of the graph must be a tree or, at least, locally tree like (that is, without short loops), (ii) the rows of the stoichiometric matrix $\mathbf{S}$ should be statistically uncorrelated. Unfortunately, neither assumption is really fulfilled by large-scale metabolic reconstructions. To give an example, consider the rows of the stoichiometric matrix for E. colicore model[24]. The rows corresponding to the adenosine–diphosphate and adenosine–triphosphate appear strongly correlated as both metabolites commonly appear in 11 reactions; the same apply for the intracellular water and hydrogen ion that have 10 reactions in common. For these reasons, MP methods suffer from all kind of convergence and accuracy problems.

In this work, we propose a new Bayesian inference strategy to analyse with unprecedented efficiency large dimensional polytopes. The use of a Bayesian framework allows us to map the original problem of sampling the feasible space of solutions of equations (1 and 2) into the inference problem of the joint distribution of metabolic fluxes. Linear and inequality constraints will be encoded within the likelihood and the prior probabilities that via Bayes theorem provide a model for the posterior $P(\boldsymbol{v}|\mathbf{b})$. The goal of this work is to determine a tractable multivariate probability density $Q(\boldsymbol{v}|\mathbf{b})$ able to accurately approximate the posterior even in the case of strongly row-correlated stoichiometric matrices. This strategy relies on an iterative and local refinement of the parameters of $Q(\boldsymbol{v}|\mathbf{b})$ that falls into the class of expectation propagation (EP) algorithms. We report results of EP for representative state-of-the-art models of metabolic networks in comparison with HR estimate, showing that EP can be used to compute marginals in a fraction of the computing time needed by HR. We also show how the technique can be efficiently adapted to incorporate the estimated growth rate of a population of Escherichia coli.

## Results

**Formulation of the problem.** We are going to formulate an iterative strategy to solve the problem of finding a multivariate probability measure over the set of fluxes $\boldsymbol{\nu}$ compatible with equations (1 and 2). For a vector of fluxes satisfying bounds 2, we can define a quadratic energy function $E(\boldsymbol{\nu})$ whose minimum(s) lies on the assignment of variables $\boldsymbol{\nu}$ satisfying the stoichiometric constraints in equation (1):

$$E(\boldsymbol{\nu}) = \frac{1}{2}(S\boldsymbol{\nu} - \mathbf{b})^T (S\boldsymbol{\nu} - \mathbf{b}) \qquad (3)$$

We define the likelihood of observing $\mathbf{b}$ given a set of fluxes $\boldsymbol{\nu}$ as a Boltzmann distribution:

$$P(\mathbf{b}|\boldsymbol{\nu}) = \left(\frac{\beta}{2\pi}\right)^{\frac{M}{2}} e^{-\frac{\beta}{2}(S\boldsymbol{\nu} - \mathbf{b})^T (S\boldsymbol{\nu} - \mathbf{b})} \qquad (4)$$

where $\beta$ is a positive parameter, the 'inverse temperature' in statistical physics jargon, that governs the penalty of whose configurations of fluxes that are far from the minimum of the energy. In a Bayesian perspective, one can consider the posterior probability of observing $P(\boldsymbol{\nu}|\mathbf{b})$ as:

$$P(\boldsymbol{\nu}|\mathbf{b}) = \frac{P(\mathbf{b}|\boldsymbol{\nu})P(\boldsymbol{\nu})}{P(\mathbf{b})} \qquad (5)$$

where the prior

$$P(\boldsymbol{\nu}) = \prod_{n=1}^{N} \psi_n(v_n) = \prod_{n=1}^{N} \frac{\mathbb{1}\left(v_n \in \left[v_n^{\inf}, v_n^{\sup}\right]\right)}{v_n^{\sup} - v_n^{\inf}} \qquad (6)$$

enforces the bounds over the allowed range of fluxes. The function $\mathbb{1}\left(v_n \in \left[v_n^{\inf}, v_n^{\sup}\right]\right)$ is an indicator function that takes value 1 if $v_n \in [v_n^{\inf}, v_n^{\sup}]$ and 0 otherwise. We finally obtain the following relation for the posterior:

$$P(\boldsymbol{\nu}|\mathbf{b}) = \frac{1}{P(\mathbf{b})} \left(\frac{\beta}{2\pi}\right)^{\frac{M}{2}} e^{-\frac{\beta}{2}(S\boldsymbol{\nu} - \mathbf{b})^T (S\boldsymbol{\nu} - \mathbf{b})} \prod_{n=1}^{N} \psi_n(v_n) \qquad (7)$$

and eventually we will investigate the $\beta \to \infty$ limit. Neglecting terms that do not depend on $\boldsymbol{\nu}$, the posterior takes the form of

$$P(\boldsymbol{\nu}|\mathbf{b}) \propto e^{-\frac{\beta}{2}(S\boldsymbol{\nu} - \mathbf{b})^T (S\boldsymbol{\nu} - \mathbf{b})} \prod_{n=1}^{N} \psi_n(v_n) \qquad (8)$$

where we have not explicitly reported the normalization constant. By marginalization of equation (7), one can determine the marginal posterior $P_n(v_n|\mathbf{b})$ for each flux $n \in \{1, \ldots, N\}$. However, performing this computation naively would require the calculation of a multiple integral that is in principle computationally very expensive and cannot be performed analytically in an efficient way.

A standard way of approximately computing $P(\boldsymbol{\nu}|\mathbf{b})$ is through sampling methods, such as the HR technique. The accuracy obtained with HR depends of course on the number of samples, and sampling accurately can be very time consuming. In the following, we develop an analytic approach to approximately compute marginal posteriors, which is able to achieve results as accurate as the HR sampling technique for a large number of sampled points in a fraction of the computing time. But first, we will describe as a warm-up naive analytic method to approximately compute marginal distributions $P_n(v_n|\mathbf{b})$.

**A non-adaptive approach.** As a first approximation, one can think of replacing each exact prior $\psi_n(v_n)$ with a single Gaussian distribution $\phi_n(v_n; a_n, d_n) = \frac{e^{-\frac{(v_n - a_n)^2}{2d_n}}}{\sqrt{2\pi d_n}}$, whose statistics, that is, the mean and the variance, are constrained to be equal to the one of $\psi_n(v_n)$. That is

$$\begin{cases} a_n &= \langle v_n \rangle_{\psi_n(v_n)} \\ d_n &= \langle v_n^2 \rangle_{\psi_n(v_n)} - \langle v_n \rangle^2_{\psi_n(v_n)} \end{cases} \quad n \in \{1, \ldots, N\} \qquad (9)$$

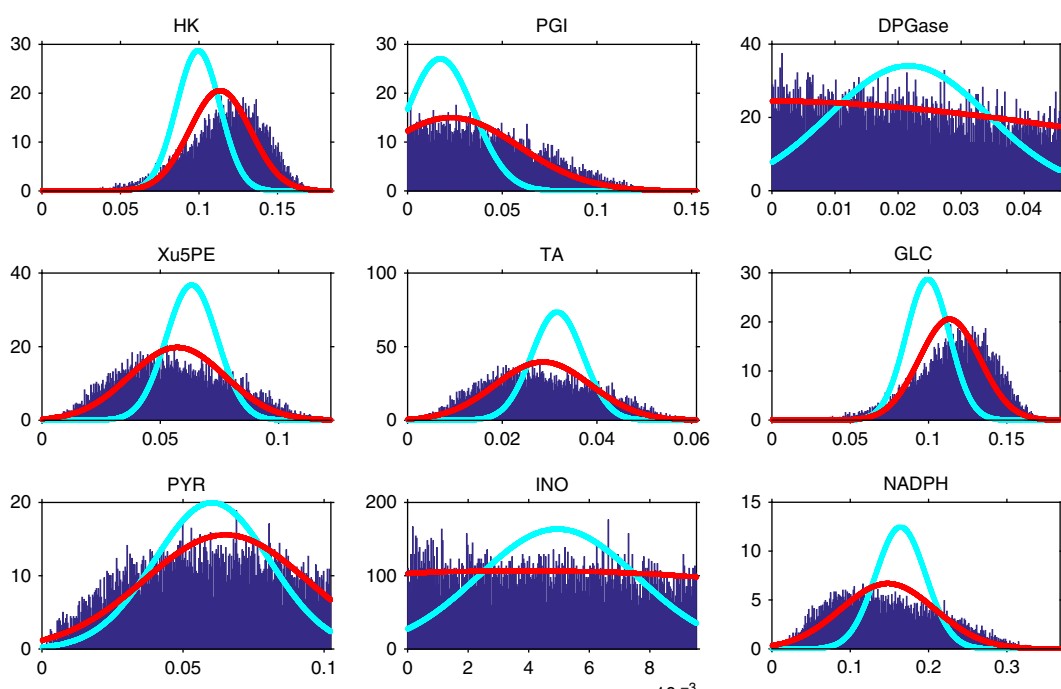

**Figure 1 | Marginal probability densities of nine fluxes of the red blood cell.** The blue bars represent the result of Monte Carlo estimate for $T \sim 10^8$ sampling points. The cyan line is the result of the non-adaptive Gaussian approximation while the red line represents the EP estimate.

We estimate the marginal posteriors from the distribution

$$Q(\boldsymbol{v}|\mathbf{b}) = \frac{1}{Z_Q} e^{-\frac{\beta}{2}(S\boldsymbol{v}-\mathbf{b})^T(S\boldsymbol{v}-\mathbf{b})} \prod_{n=1}^{N} \phi_n(v_n; a_n, d_n) \quad (10)$$

$$Z_Q = \int d^N \boldsymbol{v} \, e^{-\frac{\beta}{2}(S\boldsymbol{v}-\mathbf{b})^T(S\boldsymbol{v}-\mathbf{b})} \prod_{n=1}^{N} \phi_n(v_n; a_n, d_n) \quad (11)$$

Notice that in this approximation fluxes result unbounded. Marginals obtained by this strategy against the HR Monte Carlo estimate are shown in Fig. 1 (cyan line) for nine representative metabolic fluxes of one of the standard model for red blood cell[25]. Marginals evaluated with this simple non-adaptive strategy differ significantly from the ones evaluated with the Monte Carlo sampling technique. In the following, we will show how we can overcome this limitation by choosing different values for the means $\mathbf{a}$ and the variances $\mathbf{d}$ in equation (10) making use of the EP algorithm.

**Expectation propagation**. EP[26] is an efficient technique to approximate intractable (that is, impossible or impractical to compute analytically) posterior probabilities. EP was first introduced in the framework of statistical physics as an advanced mean-field method[27,28] and further developed for Bayesian inference problems in the seminal work of Minka[26]. Let us consider the $n$th flux and its corresponding approximate prior $\phi_n(v_n; a_n, d_n)$. We define a tilted distribution $Q^{(n)}$ as

$$Q^{(n)}(\boldsymbol{v}|\mathbf{b}) \equiv \frac{1}{Z_{Q^{(n)}}} e^{-\frac{\beta}{2}(S\boldsymbol{v}-\mathbf{b})^T(S\boldsymbol{v}-\mathbf{b})} \psi_n(v_n) \prod_{m \neq n} \phi_m(v_m) \quad (12)$$

The important difference between the tilted distribution and the multivariate Gaussian $Q(\boldsymbol{v}|\mathbf{b})$ is that all the intractable priors are approximated as Gaussian probability densities except for the $n$th prior, which is treated exactly. For this reason, we expect that this distribution will be more accurate than $Q(\boldsymbol{v}|\mathbf{b})$ regarding the estimate of the statistics of flux $n$ without significantly affecting the computation of expectations. Bearing in mind that it is a large number of exact priors (that is, the distributions $\{\psi_i\}_{i=1,\cdots,N}$) that make the computation intractable and not a single one, we have introduced only one exact intractable prior in $Q^{(n)}$.

One way of determining the unknown parameters $a_n$ and $d_n$ of $\phi_n(v_n; a_n, d_n)$ is to require that the multivariate Gaussian distribution $Q(\boldsymbol{v}|\mathbf{b})$ is as close as possible to the auxiliary distribution $Q^{(n)}(\boldsymbol{v}|\mathbf{b})$. Intuitively, there are at least two possibilities to enforce this similarity: (i) matching the first and the second moments of the two distributions (ii) minimizing the Kullback–Leibler divergence $D_{KL}(Q^n\|Q)$; these two methods coincide (see details in Supplementary Note 1). Thus, we aim at imposing the following moment matching conditions:

$$\begin{cases} \langle v_n \rangle_{Q^{(n)}} &= \langle v_n \rangle_Q \\ \langle v_n^2 \rangle_{Q^{(n)}} &= \langle v_n^2 \rangle_Q \end{cases} \quad (13)$$

from which we get a relation for the parameters $a_n$, $d_n$ that is explicitly reported in section 3.

EP consists in sequentially repeating this update step for all the other fluxes and iterate until we reach a numerical convergence. Further technical details about the convergence are reported in subsection 1. At the fixed point, we directly estimate the marginal posteriors $P_n(v_n|\mathbf{b})$, for $n \in \{1, \dots, N\}$, from marginalization of the tilted distribution $Q^{(n)}$ that turns out to be a truncated Gaussian density in the interval $[v_n^{\inf}, v_n^{\sup}]$ (see Supplementary Note 2).

At difference from the non-adaptive approach, the EP algorithm determines the approximated prior density by trying to reproduce the effect that the true prior density has on variable $v_n$, including the interaction of this term with the rest of the

system. First, the information encoded in the stoichiometric matrix is surely encompassed in the computation of the means and the variances of the approximation since both the distributions $Q^{(n)}$ and $Q$ contain the exact expression of the likelihood. Second, the refinement of each prior also depends on the parameters of all the other fluxes.

As an example of the accuracy of this technique, we report in Fig. 1 (red line) the nine best marginals estimated by EP of the red blood cell against the results of HR Monte Carlo sampling. Figure 1 suggests that this technique leads to a significant improvement of the non-adaptive approximation as the plot shows a very good overlap between the distributions provided by HR and EP. The entire set of marginals and a comparison with a state-of-the-art MP algorithm[7] is reported in the Supplementary Fig. 2.

**Numerical results for large-scale metabolic networks**. This section is devoted to compare the results of our algorithm against the outcomes of a state-of-the-art HR Monte Carlo sampling technique on three representative models of metabolic networks, precisely the *iJR904* (ref. 29), the *CHOLnorm*[30] and the *RECON1* (ref. 31) models for *E. coli*, the *Cholinergic neuron* and *Homo sapiens* organisms, respectively. In Supplementary Fig. 3, we report results for a larger set of models all selected from the Bigg models database[32].

Experiments are performed as follows. First, we preprocess the stoichiometric matrix of the model in order to remove all reactions involving metabolites that are only produced or only degraded[33].

After the preprocessing, we run HR and EP, both implemented on Matlab or as Matlab libraries, to the reduced model. Let us explain how the two methods work. Starting from a point lying on the polytope, HR iteratively chooses a random direction and collects new samples in that direction such that they also reside in the solution space. In this work, we use an optimized implementation of HR, called *optGpSampler*[6]. Regarding the HR simulations, we set the number of sampled points to be equal to $10^4$ for an increasing number of explored configurations $T$ from $10^4$ to $10^7$ in most of the cases; for some specific models, that is, very large networks having $N \sim 10^3$ reactions, we explore up to $T \sim 10^9$ points. Concerning the EP algorithm, we perform the same experiment setting the $\beta$ parameter to be equal to $10^{10}$ for almost all models. In only one case (the *RECON1* model), we encountered convergence problems and thus we decreased it to $10^9$. Numerical convergence of EP depends on the refinement of parameters $\mathbf{a}$ and $\mathbf{d}$ or, more precisely, on the estimate of the marginal distributions of fluxes. At each iteration $t$, we compute an error $\varepsilon$, which measures how the approximate marginal distributions change in two consecutive iterations. Formally, we define the error as the maximum value of the sum of the differences (in absolute values) of the mean and second moment of the marginal distribution, that is

$$\varepsilon^t = \max_n \left| \langle v_n \rangle_{Q^{(n)}}^{t+1} - \langle v_n \rangle_{Q^{(n)}}^{t} \right| + \left| \langle v_n^2 \rangle_{Q^{(n)}}^{t+1} - \langle v_n^2 \rangle_{Q^{(n)}}^{t} \right|$$

If $\varepsilon^t$ is smaller than a predetermined target precision (we used $10^{-5}$), the algorithm stops.

To quantitatively compare the two techniques, we report here the scatter plots of variances and means of the approximate marginals computed via HR and EP. Moreover, we estimate the degree of correlation among the two sets of parameters computing the Pearson product-moment correlation coefficient.

Notice that we cannot have access to the exact marginals and that we assume that the results obtained by HR are exact only asymptotically. Thus, our performances, both for the direct comparison of the means and variances and for the Pearson's

coefficient, should be considered accurate if they are approached by the Monte Carlo ones for an increasing number of explored points.

The three large subplots in Fig. 2 show the results for *E. coli*, *C. neuron* and *Homo sapiens*, respectively. For each organism, we report on the top-left panel the time spent by EP (straight line) and by HR (cyan points) and on the bottom-left panel the Pearson correlation coefficients. Both measures of time and correlation are plotted as functions of the number of configuration $T$ obtained from the HR algorithm. As shown in these plots, we can notice that to reach a high correlation regime a very large number of explored configurations, employing a computing time that is always several orders of magnitude larger than the EP running time. This is particularly strinking in the case of the *RECON1* model, for which we needed to run HR for about 20 days in order to reach results similar to the outcomes of EP, that converges in less than 1 h on the same machine.

To underline how EP seems to approach HR results in the asymptotic limit, we report in the rest of the sub-figures the scatter plots of the means (top) and the variances (bottom) of the marginals. On the $y$ axis, we plot the EP means (variances) against the HR means (variances) for an increasing number of explored configurations, as indicated in $x$ axis. Results clearly show that as $T$ grows, the points (both means and variances) are more and more aligned to the identity line: not only these measures are highly correlated for large $T$, but they assume very similar values. This is remarkably appreciable in the results for *CHOLnorm* model: for $T = 4 \times 10^4$ the means of the scatter plots are quite unaligned but as $T$ reaches $4 \times 10^7$, they almost lie on the identity line. In fact, means are poorly correlated for $T = 4 \times 10^4$ while the Pearson correlation coefficient is close to 1 for $T = 4 \times 10^7$.

**Study of *E. coli* metabolism for a constrained growth rate**. The EP formalism can efficiently deal with a slightly modified version of problem of sampling metabolic networks. Suppose to have access to experimental measurements of the distribution of some fluxes under specific environmental conditions. We would like to embed this empirical knowledge in our algorithm, by matching the posterior distribution of the measured fluxes with the empirical measurements. Within the EP scheme, this task corresponds to matching the first two moments (mean and variance) of the posteriors with the one defined by the empirical measurements. With the inclusion of empirically established prior knowledge, we want to investigate how the experimental evidence is related to the metabolism at the level of reactions or, in other words, we want to determine how fluxes modify in order to reproduce the experiments. In this perspective, the EP scheme can easily accommodate additional constraints on the posteriors by modifying the EP update equations as outlined in Methods section.

We have tested this variant of EP algorithm on the *iJR904* model of *E. coli* for a constrained growth rate. In fact, one of the few fluxes that are experimentally accessible is the biomass flux, often measured in terms of doubling per hour. As a matter of example, we decide to extract one of the growth rates reported in Fig. 3a of ref. 34; the profile labelled as *Glc (P5-ori)* can be well fitted by a Gaussian probability density of mean $0.92 \, \text{h}^{-1}$ and variance $0.0324 \, \text{h}^{-2}$. This curve represent single-cell measures of a population of bacteria living in the so-called minimal substrate whose main characteristics are in principle well caught by the *iJR904* model. We fixed the bound on the glucose exchange flux EX_glc(e) such that the maximum allowed growth rate (about $2 \, \text{h}^{-1}$) contained all experimental values in the profile labelled as *Glc (P5-ori)* of Fig. 3a of ref. 34. This was easily computed by

fixing the biomass flux to the desired value and minimizing the glucose exchange flux using FBA, and gies a the lower bound of the exchanged glucose flux of $-43 \, \text{mmol} \, (\text{g[DW]})^{-1} \, \text{h}^{-1}$.

We then apply EP algorithm to the modified *iJR904* model in two different conditions. First, we do not impose any additional constraint and we run the original EP algorithm as described in the previous section. Then, as described in Methods section, we fix the marginal posterior of the biomass. We can now compare the means and the variances of all the other fluxes in the two cases and single out those fluxes that have been more affected by the empirical constraints on the growth rate. We report in Fig. 3 the plot of the ratio between the means (Fig. 3a) and the variances (Fig. 3b) in the unconstrained case and in the constrained case. In Fig. 3a, these ratios are plotted against the logarithm of the absolute value of the unconstrained means to differentiate those fluxes having means close to zero and all the other cases. The ratios of the variances are instead plotted as a function of the unconstrained variances in semi-log scale. We can notice that apparently a large fraction of the fluxes have changed their marginal distribution in order to accommodate the fixed marginal for the biomass. We have reported the name of the reactions with the most significant changes; for instance, the marginal of the TKT2 reaction has reduced its mean of more than one third, while many reactions involving aspartate have significantly lowered their variances.

To underline the non-trivial results of EP algorithm in the constrained case, we apply again the standard EP algorithm to the *iJR904* model when the lower bound and the upper bound of the biomass is fixed to the average value of the experimental profile. The comparison (not shown) between the two approaches suggests that the most relevant change concerns the *EX_asp_L(e)* flux as both the average value and the variance estimated in the second case are about two times the ones predicted by constrained EP. The distributions of most other fluxes remain do not considerably change. We underline that the different behaviour of the marginals in the two cases, even if not significant for most of the fluxes, was in principle unpredictable without the use of constrained EP; and we do not exclude that fixing other empirical profiles can lead to very different results. Likewise, it seems unlikely that the results computed with constrained EP could be obtained using unbiased samples as provided by standard HR implementations (see a discussion in Supplementary Note 6).

## Discussion
In this work, we have shown how to study the space of feasible configurations of metabolic fluxes within a cell via an analytic description of the marginal probability distribution characterizing each flux. Such marginals are described as truncated Gaussian densities whose parameters are determined through an iterative and extremely efficient algorithm, the EP algorithm. We have compared our predictions against the estimates provided by HR sampling technique and results shown in Subsection 1 suggest a very good agreement between EP and HR for a large number of explored configurations, $T$. First of all, the direct comparison of the means and variances of EP versus HR reported in the scatter plots shows that the more we increment the HR points, the more the scatter points are aligned. Second, we see an increment of the correlation between EP and HR statistics for an increasing number of sampled points; correlations reach values very close to 1 for large values of $T$ and for almost all the models we have considered. The most important point is that the computation times of EP, at high correlation regime, are always orders of magnitude lower than HR sampling times. This is extremely time-saving when we deal with very large networks, as the

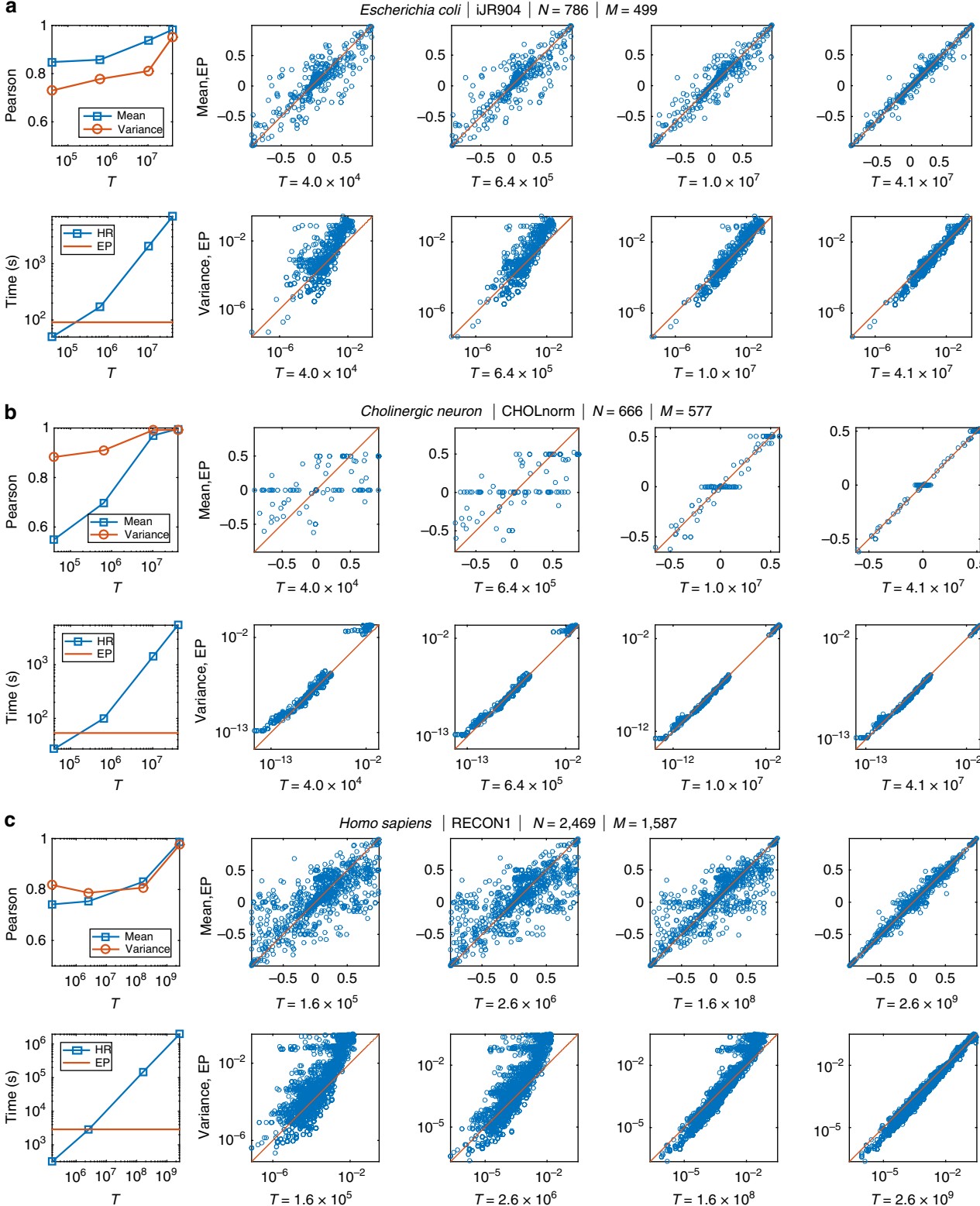

**Figure 2 | Comparison of the results of HR versus EP.** HR vs. EP prediction for three large-scale metabolic reconstructions: iJR904 (**a**), CHOLnorm (**b**), and RECON1 models (**c**). The top-left plot shows the Pearson correlation coefficients between variances and means estimated through EP and HR. The bottom-left panel reports the computing time of EP and HR for different values of $T$. The plots on the right are scatter plots of the means and variances of the approximated marginals computed via EP against the ones estimated via HR for an increasing number of explored configurations $T$.

*RECON1* model for *Homo sapiens* where the running time (in seconds) of EP is three order of magnitude smaller than HR. We underline that exact marginals are generally inaccessible and we cannot compare our results against a ground-truth; our measures

well approximate 'true' distributions as long as the exactness of HR in the asymptotic limit is correct.

We have shown how to include empirical knowledge on distribution of fluxes on the EP algorithm without compromising

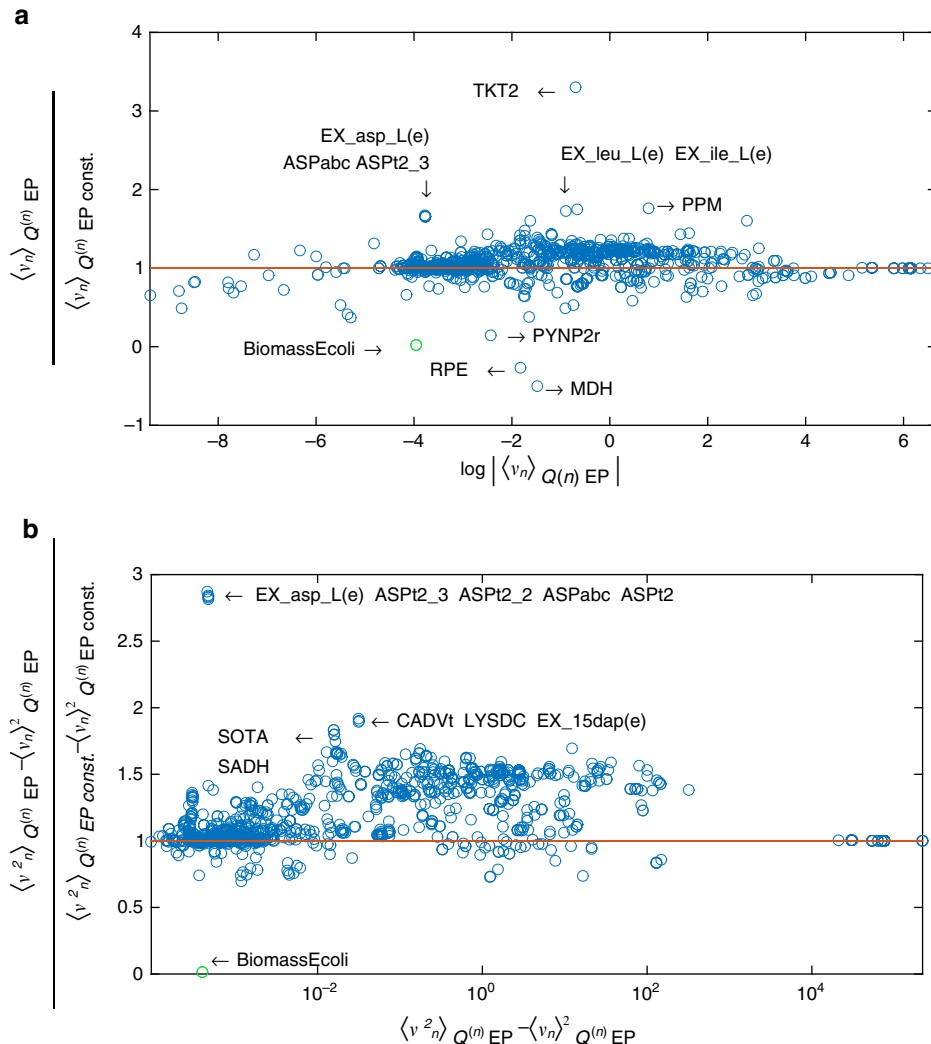

**Figure 3 | Results for a constrained biomass flux.** Comparison between the means (**a**) and variances (**b**) of the marginal probability densities for all the fluxes computed without the additional constraint (unconstrained case) and with the constrained on the biomass (constrained case). The green point indicates the biomass flux.

the computing time. More precisely, we have investigated how fixing an experimental profile of the growth rate into the *iJR904* model of *E. coli* affect non-trivially all other fluxes. This is a remarkable advantage of the EP algorithm with respect to other methods.

EP provides an analytic estimate of each single flux marginal, which relies on the optimization of two parameters, the mean and the variance of a Gaussian distribution. The formalism allows in principle more complicated parametrizations of posteriors to include other biological insights.

EP equations are extremely easy to derive and to implement, as the main loop can be written in few lines of Matlab code. The method is iterative, and the number of operations in each iteration scales as $\Theta(N^3)$, rendering EP extremely convenient in terms of computation time with respect to existing alternatives.

An shown in Fig. 2 in real cases, variances of the marginal distributions can span several orders of magnitude. This range of variability implies that also the variances of the approximation need to allow both very small and huge values. To cope with the numeric problems that may arise, we allow parameters $d$ to vary in a finite range of values with the drawback of limiting the set of allowed Gaussian densities of the approximation. For instance, a flat distribution cannot be perfectly approximated through a

Gaussian whose variance cannot be arbitrary large; in the opposite extreme, imposing a lower bound on variances prevents the approximation of posteriors that are too concentrated on a single point. Thus, this range needs to be reasonably designed in order to catch as many 'true' variances as possible. In this work, we have tried to impose a very large range of values, typically $(10^{-50}, 10^{50})$, to include as many distributions as possible without compromising the convergence of the algorithm. Moreover, the Gaussian profile itself is surely a limitation of the approximation as true marginals can have in principle arbitrary profiles.

EP performances are sensitive to the parameter $\beta$ and equations become numerically unstable for too large $\beta$ (for example, $10^{11}$–$10^{12}$). On the other hand, $\beta$ can be seen as the inverse-variance of a Gaussian noise affecting the conservation laws. The nature of this noise could depend on localization properties on the cell and real thermal noise. In this case, an optimization of the free energy with respect to $\beta$ can in principle lead to better predictions.

## Methods
**Update rule**. The algorithm described in the EP section relies on local moves in which, at each step, we refine only the parameters of one single prior, minimizing

the dissimilarity between the auxiliary tilted distribution $Q^{(n)}$ and $Q$. The values of the mean and of the variance of $\phi_n(v_n; a_n, d_n)$ are iteratively tuned in a way that the first and second moments of the two distributions match. The update rule for the parameters $a_n$ and $d_n$ of the Gaussian prior will be derived in details in the following section.

Let us express the auxiliary density $Q^{(n)}$ in equation (12) as a standard multivariate Gaussian distribution times the exact prior of the $n$th flux as

$$Q^{(n)}(\boldsymbol{\nu}|\mathbf{b}) = \frac{1}{Z_{Q^{(n)}}} e^{-\frac{\beta}{2}(S\boldsymbol{\nu}-\mathbf{b})^T(S\boldsymbol{\nu}-\mathbf{b})-\frac{1}{2}(\boldsymbol{\nu}-\mathbf{a})^T D(\boldsymbol{\nu}-\mathbf{a})} \psi_n(v_n) \quad (14)$$

$$= \frac{1}{\tilde{Z}_{Q^{(n)}}} e^{-\frac{1}{2}(\boldsymbol{\nu}-\bar{\boldsymbol{\nu}})^T \Sigma^{-1}(\boldsymbol{\nu}-\bar{\boldsymbol{\nu}})} \psi_n(v_n) \quad (15)$$

where $\tilde{Z}_{Q^{(n)}} = Z_{Q^{(n)}} e^{\frac{\beta}{2}\mathbf{b}^T\mathbf{b}-\frac{1}{2}\bar{\boldsymbol{\nu}}^T\bar{\boldsymbol{\nu}}}$, $D$ is a diagonal matrix with components $D_{mm} = \frac{1}{d_m}$ for $m \neq n$ and $D_{nn} = 0$ (and of course non-diagonal terms $D_{mk} = 0$ if $m \neq k$). The covariance matrix $\Sigma^{-1}$ and the mean vector $\bar{\boldsymbol{\nu}}$ satisfy:

$$\begin{cases} \Sigma^{-1} & = & \beta S^T S + D \\ \bar{\boldsymbol{\nu}} & = & \Sigma(\beta S^t \mathbf{b} + Da) \end{cases} \quad (16)$$

Note that we are omitting for notational simplicity the dependence of $D, \Sigma, \bar{\boldsymbol{\nu}}$ on $n$. Equivalently

$$Q(\boldsymbol{\nu}|\mathbf{b}) = \frac{1}{\tilde{Z}_Q} e^{-\frac{1}{2}(\boldsymbol{\nu}-\bar{\boldsymbol{\nu}})^T \Sigma^{-1}(\boldsymbol{\nu}-\bar{\boldsymbol{\nu}})} \phi_n(v_n; a_n, d_n) \quad (17)$$

where $\tilde{Z}_Q = Z_Q e^{\frac{\beta}{2}\mathbf{b}^T\mathbf{b}-\frac{1}{2}\bar{\boldsymbol{\nu}}^T\bar{\boldsymbol{\nu}}}$. If we now exploit the moment matching condition in equation (13) (a detailed calculation of the moments of $Q$ and $Q^{(n)}$ expressed in standard form is reported in Supplementary Notes 2 and 3) we obtain an update equation for the mean and the variance:

$$\begin{cases} d_n & = & \left(\frac{1}{\langle v_n^2 \rangle_{Q^{(n)}} - \langle v_n \rangle_{Q^{(n)}}^2} - \frac{1}{\Sigma_{nn}}\right)^{-1} \\ a_n & = & d_n\left[\langle v_n \rangle_{Q^{(n)}}\left(\frac{1}{d_n} + \frac{1}{\Sigma_{nn}}\right) - \frac{\bar{v}_n}{\Sigma_{nn}}\right] \end{cases} \quad (18)$$

Notice that the sequential update scheme described in the EP section requires the inversion of the matrix $\Sigma^{-1}$ each time that we have to refine the parameters of flux $n$, leading to $N$ inversions per iteration amounting to $\Theta(N^4)$ operations per iteration. We propose in Supplementary Note 4 a parallel update that needs only one matrix inversion per iteration, that is, $\Theta(N^3)$ operations per iteration.

**Update equations for a constrained posterior.** Let us assume to have access to experimental measures of the (marginal) posterior $f(v_i)$ for flux $i$. We aim at determining how the posteriors of other fluxes would modify to fit with the experimental results compared, for instance, to the unconstrained case. The so-called maximum entropy principle[35] dictates that the most unconstrained distribution which is consistent with the experiment, prior distributions and flux conservation $S\boldsymbol{\nu} = \mathbf{b}$, is simply

$$P(\boldsymbol{\nu}|\mathbf{b}) = \frac{1}{Z} e^{-\frac{\beta}{2}(S\boldsymbol{\nu}-\mathbf{b})^T(S\boldsymbol{\nu}-\mathbf{b})} \prod_{n=1}^{N} \mathbb{1}\left(v_n \in [v_n^{\inf}, v_n^{\sup}]\right) g(v_i) \quad (19)$$

where $\beta \to \infty$ and $g(v_i)$ is the (exponential of the) function of unknown Lagrange multipliers that has to be determined in order for the constraint $\int \prod_{n \neq i} dv_n P(\boldsymbol{\nu}|\mathbf{b}) = f(v_i)$ to be satisfied. In the particular case in which the posterior can be reasonably fitted by a Gaussian distribution $\mathcal{N}(v_i|a_i^{\exp}, d_i^{\exp})$, then it suffices to consider also a Gaussian $g(v_i) = \mathcal{N}(v_i|a_i, d_i) = \phi_i(v_i|a_i, d_i)$ with only two free parameters. The determination of $a_i, d_i$ can be achieved by slightly modifying the EP update for flux $i$. Assuming as before that the prior of each flux $n \neq i$ can be approximated as a Gaussian profile $\phi_n(v_n; a_n, d_n)$ of parameters $a_n$ and $d_n$, also to be determined, we must impose that

$$\mathcal{N}(v_i|a_i^{\exp}, d_i^{\exp}) \propto \mathcal{N}(v_i|a_i, d_i) \int \prod_{n \neq i} dv_n Q(\boldsymbol{\nu}|\mathbf{b}) \quad (20)$$

$$\propto \phi_i(v_i; a_i, d_i) e^{-\frac{(v_i - \bar{v}_i)^2}{2\Sigma_{ii}}} \quad (21)$$

where the distribution $Q(\boldsymbol{\nu}|\mathbf{b})$ is the one in equation (10). Since both the left-hand side and the right-hand side of equation (21) contain Gaussian distributions, the relations for $a_i$ and $d_i$ can be easily computed and take the form

$$\begin{cases} d_i & = & \left(\frac{1}{d_i^{\exp}} - \frac{1}{\Sigma_{ii}}\right)^{-1} \\ a_i & = & d_i\left(\frac{a_i^{\exp}}{d_i^{\exp}} - \frac{\bar{v}_i}{\Sigma_{ii}}\right) \end{cases} \quad (22)$$

This expression is exactly the same in equation (18) if we replace the mean and the variance of the tilted distribution with the experimental ones.

**Technical details.** The computations were performed on a Dell Poweredge server with 128 Gb of memory and 48 AMD Opteron CPUs clocked at 1.9 Ghz. No constraint have been placed on the number of CPU threads, allowing both EP and HR to parallelize their processes. We observed that EP used 2–3 cores, exclusively in the matrix inversion phase (which was time-dominant), while HR employed a variable number of cores (around six or seven at some times). For this reason only, the order of magnitude of computing times of HR and EP are fairly comparable but they are sufficient to underline the differences between the two algorithms.

**Code availability.** An implementation of the algorithm presented in this work is publicly available at https://github.com/anna-pa-m/Metabolic-EP.

**Data availability.** All data generated or analysed during this study are included in the manuscript and its Supplementary Information file.

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

## Acknowledgements

We warmly thank Andrea De Martino, Roberto Mulet, Jorge Fernández de Cossio and Eduardo Martinez Montes for interesting discussions. We acknowledge support from project SIBYL, financed by Fondazione CRT under the initiative 'La ricerca dei Talenti' and project 'from cellulose to biofuel through Clostridium cellulovorans: an alternative biorefinery approach' of University of Turin, financed by Compagnia di Sanpaolo.

## Authors contributions

All authors contributed equally to this work.

## Additional information

**Competing interests:** The authors declare no competing financial interests.

**Publisher's note**: 

