## [Peer review file · Nature Communications]

Reviewers' comments:

Reviewer #1 (Remarks to the Author):

The paper proposes a transformation of the sampling of feasible steady-state flux distributions into an inference problem of the joint distribution of metabolic fluxes. The inference problem is efficiently solved via an iterative algorithm which falls in the class of Expectation Propagation algorithms. The performance of the algorithm is illustrated on several large-scale metabolic networks from different organisms and demonstrates good correspondence between the first and second moment of individual fluxes. However, there are several key issues which need further attention, some of them related to the structure and writing and others focusing on the proposed method itself.

Structure and writing

1. Lines 11 to 19 of the abstract are standard in the metabolic community and can be summarized in a single sentence. It is not clear what the authors mean by “reference tools” since the enumerated serve to solve different problems. Moreover, the example for coupling of fluxes is not the prime one; in fact, coupling of fluxes (in the sense that their ratio is the same in any steady state of the system) serves as a more intuitive example. What does the acronym TAP stand for (this is not clear for the general audience of Nature communications)? Due to these issues, a more informative and to-the-point abstract is needed.
2. In section I, line 54 – 55, it is not clear what is meant by “if the optimum is unique”? A linear program has a unique optimum value, but the optimizer (the set of values for the variables) may not be.
3. It is not clear why incorporation of prior knowledge (e.g. estimated fluxes) is limiting to the HR technique (lines 73 - 77).
4. It would be nice to provide biochemical examples when the rows of the stoichiometric matrix would be correlated? (e.g. cofactors usage)
5. Line 124 can include a reference to Eq. (II.8).
6. Metabolites do not carry flux (line 172). I suppose metabolites which are only produced or only degraded are removed.
7. The technical details of the implementation can be postponed from lines 180 – 183 to the methods section.
8. The statement on lines 190 – 191 is obsolete.
9. What is Cholinergic in Fig. II.2? The Figure can be reduced to only the leftmost panels, describing the important message and the paragraph on lines 195 – 207 should be rewritten. Most importantly, how do the authors explain the observation that there is a part in the space of possible second moments in the flux space which is not populated (as seen in Fig. II.2 right panels)?
10. The discussion deserves more rigor and more biological examples to demonstrate the usefulness of the approach (apart from lowering the computation time for similar approximation of marginals). What do the authors mean by sufficient statistics of a Gaussian distribution? In what sense are the $N \times N$ and $M \times M$ matrices equivalent?
11. Parts of section IV.B cannot be understood without reading the appendices as clarification of the notation used.

Comments to the Method

1. Why is the normalization constant from Eq. II.2 neglected if the limits of II.5 in limits of β to infinity?
2. It seems that the fluxes shown in Figure II.1 involve exchange reactions. How does the first local approach behave for internal reactions?
3. What are the rules for numerical convergence used in this study (lines 151 - 152).
4. Doesn't Figure II.1 (b) report the accuracy rather than the efficiency of the approach?
5. Most critically, how does one choose the value of β a priori? Is there any guideline for this important issue? If not resolved, the approach will have the same issues as the criticized HR whose convergence depends on the scenarios investigated (as stated in the introduction).
6. The error in the means and the variances should be a function of a and d , the parameters to be estimated (see lines 178).
7. The discussion is convoluted when it comes to the treatment of the ranges for b allowed! Why is a range from 10^{-50} to 10^{50} considered and how is this precision achieved in the implementation? It is clear that a simple variant of Flux Variability Analysis can determine the range of values that a flux is allowed to take in the feasible space. Why not use these as bounds?
8. The authors should provide one biologically motivated example where experimental evidence is encoded (e.g. confidence intervals of flux estimates from labeling studies in different model species).
9. The definition of D does not match the standard definition of a diagonal matrix!
10. The S used for stoichiometric matrix in the introduction is different from that in Eq. IV.3. Please, unify.
11. Are there any fluxes for which convergence is achieved easier than for others?
12. It is desirable to include a detailed discussion about the assumption $d_n \neq 0$ in line 366 (to derive Eqs. A.11 and A.12), especially in light of including additional constraints (which render it more likely that this condition is violated).

Reviewer #2 report of NCOMMS-16-10245 entitled An analytic approximation of the feasible space of metabolic network by Braunstin, Muntoni, Pagnani

Summary of the Paper

In this work, the authors introduce a novel method to estimate the solution space for the flux vector of a set of coupled chemical reactions assumed to be in the stationary state. The method goes further than previous methods based on techniques coming from the area of disordered systems and that go generally under the name of belief propagation algorithms, cavity equations, and so on. The authors claim correctly that their new method, called Expectation Propagation, is very simple and iterative, and overcomes previous known problems found in other algorithms (Hit-and-Run (HR) algorithm, belief propagation, etc). They substantiate their claims with extensive numerical analysis and comparison with HR algorithm, which is taken as providing the numerically exact result in the metabolic networks under study.

Although the method of expectation propagation was introduced some time ago by Minka, the author's results, applications and adaptations of such method to the estimation of reaction fluxes are indeed novel and I strongly believe that there will be of interest in the scientific community, as well as in other research fields in which similar mathematical problems arise. Moreover, I hope that biologists still using Flux Balance Analysis -and modified techniques- will be attracted into using this new tool and, therefore, their methodology can be greatly influenced.

The numerical benchmarks presented in the paper seem clear to me, and convey correctly the message that the authors intend to deliver, namely, the mathematical power of the method exposed in the paper.

Therefore I recommend the paper for publication in nature communications. I have, however, a lists of major and minor comments the authors must address first.

Major and minor comments

1. The technique of Flux Balance Analysis should be mentioned explicitly in the abstract (not just as linear programming).
2. Line 25: Spell out TAP
3. Line 27: Cavity \rightarrow cavity
4. Line 47: $\nu,^{inf} \nu^{sup} \rightarrow \nu^{inf}, \nu^{sup}$
5. Line 49: After the sentence "Empirically, it turns out that $N \gg M$ so that the system is typically severely undetermined", it should be perhaps appropriate to give some examples of values N and M for some metabolic networks, to give an idea to the reader.
6. Line 52-53. I wonder whether the author agree with me in that the following reference could be useful for some readers regarding Flux Balance Analysis:
 - * Kauffman, Kenneth J., Purusharth Prakash, and Jeremy S. Edwards. "Advances in flux balance analysis." *Current opinion in biotechnology* 14.5 (2003): 491-496.
7. Line 64: "Basically consists" \rightarrow "Basically, it consists"
8. Line 79: The following references seem to be missing (importantly, the second one):
 - * Font-Clos, Francesc, Francesco Alessandro Massucci, and Isaac Pérez Castillo. "A weighted belief-propagation algorithm for estimating volume-related properties of random polytopes." *Journal of Statistical Mechanics: Theory and Experiment* 2012.11 (2012): P11003.
 - * Fernandez-de-Cossio-Diaz, Jorge, and Roberto Mulet. "Fast inference of ill-posed problems within a convex space." *arXiv preprint arXiv:1602.08412* (2016).
9. In the section "Formulation of the problem" (from line 102), I have the following suggestions/comments:

- (a) I think that the authors should somewhat clarify early how the bounds on the fluxes is taken into account. I am worried that a reader, might start wondering about this from line 105 forward. Perhaps, the following rephrase on line 104 might be useful: "As standard, and ignoring for the moment the bounds in the fluxes, we can define a quadratic energy function...", or something similar.

10. In the section "First local approach" (from line 121), I have the following suggestions/comments:

- (a) Although it is implicit in the form of the Gaussian prior (line 123), it should be made explicit that, with this choice, the fluxes are now unbounded.
- (b) In Eq. (II.8) the symbol $\forall n$ is used, while in line 114 the symbol $n \in \{1, \dots, N\}$ is used. Please, unify presentation of the notation.

11. In the section "Expectation Propagation" (from line 131), I have the following suggestions/comments:

- (a) Line 132: "Expectation Propagation [19]" \rightarrow "Expectation Propagation (EP) [19]"
- (b) Eq (II.9) is not clear. A better alternative is:

$$Q^{(n)}(\boldsymbol{\nu}|\mathbf{b}) \doteq \frac{1}{Z_{Q^{(n)}}} e^{-\frac{\beta}{2}(\mathbf{S}\boldsymbol{\nu}-\mathbf{b})^T(\mathbf{S}\boldsymbol{\nu}-\mathbf{b})} \psi_n(\nu_n) \prod_{m \neq n} \phi_m(\nu_m) \quad (1)$$

- (c) Moreover the sign \doteq has various meanings in Mathematics, and for some readers it may not be clear from the context.
- (d) In reference [24] of line 143, it is said: "Of course, it is an extensive number of "intractable" priors that makes the computation intractable, not a single one!". Does this mean that a possible generalization of the EP method would be to include an intensive number of intractable priors, viz

$$Q^{(n_1, \dots, n_L)}(\boldsymbol{\nu}|\mathbf{b}) \doteq \frac{1}{Z_{Q^{(n)}}} e^{-\frac{\beta}{2}(\mathbf{S}\boldsymbol{\nu}-\mathbf{b})^T(\mathbf{S}\boldsymbol{\nu}-\mathbf{b})} \psi_{n_1}(\nu_{n_1}) \cdots \psi_{n_L}(\nu_{n_L}) \prod_{m(\neq n_1, \dots, n_L)} \phi_m(\nu_m), \quad ? \quad (2)$$

According to your expertise, how will this affect the viability of your method for moderate values of L ? I guess that it must be important which set of fluxes (n_1, \dots, n_L) are chosen in this generalization (ie. highly correlated fluxes)

- (e) The panels (a) and (b) of figure II.1 should be made bigger (space permitting) and the labels in the axes also bigger, so it improves readability.
- (f) If we take e.g. $L = 2$ in Eq. (2) of this report and take appropriate choice of pairs of fluxes, could we further improve the results of panel (b) in figure II.1.
- (g) On panel (b) of figure II.1 the authors compare the results of EP with HR. However, I believe they should also compare with what it is the state-of-the-art of BP algorithms presently, which is the one presented in Fernandez-de-Cossio-Diaz, Jorge, and Roberto Mulet. "Fast inference of ill-posed problems within a convex space." arXiv preprint arXiv:1602.08412 (2016).

12. In the subsection "Numerical results" (from line 163)

- (a) The order in the discussion between lines 168-172 is a bit confusing. Perhaps it would be more appropriate to discuss the part regarding the preprocessing the network first and then discuss how HR and EP are run in the reduced networks.
- (b) In line 182: with MS, do you mean EP?
- (c) In line 193: "Person's" \rightarrow "Pearson's"
- (d) In line 190-191: "This coefficient is equals..." \rightarrow "This coefficient is equal..."
- (e) In line 195: "Per each" \rightarrow "For each"
- (f) In line 202: "Results suggest that the more T grows the more the points, either the means or the variances, are correlated" \rightarrow "Results suggest that as T grows, more points, either the means or the variances, become correlated"

13. In the section "Discussions" (from line 209)

- (a) In line 209 "Discussions" \rightarrow "Discussion".

- (b) In line 236, the power EP algorithm is mentioned for the first time. Is this a new implementation? Is this a known method? If the latter, can a reference be provided?. This paragraph does not seem to add anything substantial to the paper. If the authors agree, perhaps it should be removed altogether from the paper (and the corresponding the appendix G). Otherwise, please give a some argument about the importance of such a variation.

14. In the section "Methods" (from line 260)

- (a) I am not sure the concept of exclusive and inclusive KL divergence is known. Please make sure this is explained in the references provided or explain otherwise.

Response to reviewers

Dear Editors,

We acknowledge both Reviewers for their critical reading of the manuscript. Thanks to their comments, we hope to have improved substantially the quality in this revised version.

In the following we list the major changes relative to the first version of the work:

A non-adaptive approach

- In the section “A non-adaptive approach” (called “First local approach” in the old submission) of the manuscript we have decided to study another organism, the red blood cell, and to report the plots of a subset of marginals. We believe that (a) the comparison to this model is very popular as many authors choose to report their results on this model (b) Reviewer 2 asked to compare some marginals estimated by EP against the BP algorithm in

* *Fernandez-de-Cossio-Diaz, Jorge, and Roberto Mulet. "Fast inference of ill-posed problems within a convex space." arXiv preprint arXiv:1602.08412 (2016).*

Since authors of this work reported the entire set of marginals for the red blood cell, we have decided to analyze the same organism and to make a comparison with their published results. The plots are shown in Appendix H.

Expectation Propagation

- The old section “Numeric results”, now entitled “Numerical results for large scale metabolic networks” , has been almost entirely rewritten to clarify several issues reported by the reviewers along with to correct some minor errors of the first submission.
- In the new section “Study of Escherichia Coli metabolism for a constrained growth rate” we report the results of the Expectation Propagation algorithm applied to the iJR904 when we constrain the biomass flux of Escherichia Coli to be equal to an experimental profile. In the first submission we just mentioned how to cope, in principle, this slightly different problem involving metabolic fluxes.

Methods

- The update rules for the experimentally measured flux are now reported in the new section “Update equations for a constrained posterior” and not anymore in the Appendix.
- The technical details of the computations are now listed in the section “Technical details”.
- All the discussion about the link between EP and other inference techniques, along with the free energy functional, has been postponed in the Appendix.

Appendix

- We have added a new section “Weighted Hit-and-Run” where we propose a simple re-weighting of the configurations sampled by HR in order to “force” a certain marginal to fit an experimental one (as we have exactly done by EP in the section “Study of Escherichia Coli metabolism for a constrained growth rate”). We have discussed in detail why this method is generally unfeasible.
- We have erased the section “Power EP” as the algorithm presented in this part has never been used for the results reported in the manuscript and thus the description of “Power EP” algorithm does not add any relevant information to the article.

Furthermore, we have applied these minor changes to the work:

- Eq. II.4. We have defined the indicator function \mathbb{I} ;

- In Appendix E, matrix \mathbf{S} was erroneously indicated as \mathbf{K} ; we have replaced all the “ \mathbf{K} ” with the correct notation;
- Generally, we have used a more unified notation for mathematical symbols. We have used lowercase bold style for vectors, uppercase bold for matrices and the apex “ T ” for matrix transposes;
- Line 399. We have changed the title of Appendix E “Fast computation of Σ and μ ” with the more appropriate “Fast computation of Σ and $\bar{\nu}$ ”;
- Section II.B is now titled “A non-adaptive approach” instead of “First local approach”;

Minor changes

- We implemented several minor changes targetted at having a much more uniform notation.
- Line 196. “the name of the organism, the name” becomes “the name of the organism”

RESPONSE TO REVIEWER 1

Structure and writing

- Q1: *Lines 11 to 19 of the abstract are standard in the metabolic community and can be summarized in a single sentence. It is not clear what the authors mean by “reference tools” since the enumerated serve to solve different problems. Moreover, the example for coupling of fluxes is not the prime one; in fact, coupling of fluxes (in the sense that their ratio is the same in any steady state of the system) serves as a more intuitive example. What does the acronym TAP stand for (this is not clear for the general audience of Nature communications)? Due to these issues, a more informative and to-the-point abstract is needed.*

A1: We shortened lines 11-19 to be hopefully more concise and to the point. We clarified that the “reference tools” depend on the type of analysis sought. We clarified a bit the example on coupling of fluxes, but kept our version as it is more generic. We were afraid that introducing the example proposed by the referee, even if much simpler to understand, would trivialize the issue, as it can be “cured” very simply by eliminating one of the two fluxes. We removed TAP from the abstract, as it is not strictly needed.

- Q2: *In section I, line 54 – 55, it is not clear what is meant by “if the optimum is unique”? A linear program has a unique optimum value, but the optimizer (the set of values for the variables) may not be.*

A2: We simplified the sentence, as the optimizer set was referenced before.

- Q3: *It is not clear why incorporation of prior knowledge (e.g. estimated fluxes) is limiting to the HR technique (lines 73 - 77).*

A3: With “prior knowledge” we wanted to indicate the inclusion of experimentally measured marginal probabilities of fluxes. We have explained in Appendix G how the solution of this problem can be attempted both by importance sampling in a Boltzmann learning scheme or by a re-weighting of the configurations explored by HR. However this last possibility turns out to be unfeasible in general cases (we suspect that the first possibility would be also very time consuming, but it seems that such tools are not available for testing currently). We have changed that part of the sentence to “incorporation of other constraints”.

- Q4: *It would be nice to provide biochemical examples when the rows of the stoichiometric matrix would be correlated? (e.g. cofactors usage)*

A4: Indeed, the suggestion is correct. We added some specific examples from the stoichiometric matrix of the *ecoli-core* model. We have added in the introduction the following lines: “To give an example, let us analyze the rows of the stoichiometric matrix for *ecoli-core* model. The rows corresponding to the adenosine-diphosphate (ADP) and adenosine-triphosphate (ATP) appear strongly correlated as both metabolites commonly appear in 11 reactions; the same apply for the intracellular water and hydrogen ion that have 10 reactions in common.”

- Q5: *Line 124 can include a reference to Eq. (II.8).*

A5: We have changed the order of Eq. (II.8) and Eqs. (II.6), (II.7) in a way that now Eq. (II.8) results immediately after Line 124 where we have explained how to determine the means and variances in the non-adaptive approach (called “first local approach” in the first submission).

- Q6: *Metabolites do not carry flux (line 172). I suppose metabolites which are only produced or only degraded are removed.*

A6: Exactly. We have clarified the sentence as “First we preprocess the stoichiometric matrix of the model in order to remove all reactions involving metabolites that are only produced or only degraded. Consider, for instance, the i^{th} metabolite entering in $K = \{k_1, \dots, k_K\}$ reactions. If the entries of the i^{th} rows of the stoichiometric matrix $\{S_{ik}\}_{k \in K}$ are all positive or all negative and moreover $b_i = 0$, the linear equation $\sum_{k=k_1}^{k_K} S_{ik}\nu_k = 0$ has only one trivial solution $\nu_k = 0$ for $k \in K$. As this assignment should be compatible with the constraint on the bounds, i.e. $\nu_k^{inf} \leq 0 \leq \nu_k^{sup}$ for $k \in K$ (otherwise the model is unfeasible), these fluxes can be set to zero slightly simplifying the model.”

- Q7: *The technical details of the implementation can be postponed from lines 180 – 183 to the methods section.*

A7: We have created a new subsection in the “Methods” part entitled “Technical details” containing the paragraph in objective.

- Q8: *The statement on lines 190 – 191 is obsolete.*

A8: Following the suggestion of the referee, we have removed the statement.

- Q9: *What is Cholinergic in Fig. II.2? The Figure can be reduced to only the leftmost panels, describing the important message and the paragraph on lines 195 – 207 should be rewritten. Most importantly, how do the authors explain the observation that there is a part in the space of possible second moments in the flux space which is not populated (as seen in Fig. II.2 right panels)?*

A9: With “Cholinergic” we refer to the metabolic reconstruction in the Bigg Models database [35] of the Cholinergic neuron, a nerve cell which uses mostly the neurotransmitter acetylcholine to send its messages.

Although we understand that the main result can be summarized in the left panel of Fig. II.2, we believe that the scatter plots are useful to describe in detail how EP results approach the Monte Carlo ones for an increasing number of sample points. Not only the means and variances of our approximation are highly correlated with HR ones, but they approach the values computed with HR as we sample the polytope more and more fairly. Moreover, the plot shows that there are outliers in some cases, and information that is difficult to infer from the correlation alone in particular when the agreement is not so good (see e.g. the figure corresponding to *Saccharomyces Cerevisiae* iND750 or *Mycobacterium Tuberculosis* iNJ661 in the Appendix).

The part in lines [195 - 207] have been rewritten.

The populated portion of the variances of fluxes for Cholinergic neuron lie on the extremes of the plot. We do not know why the middle region is not populated. The fluxes with smaller variances ($< 10^{-10}$) are essentially blocked as their mean is also close to zero. There is however a consistent group of fluxes with small variances ($< 10^{-5}$) but non-zero means; that is fluxes for which there is essentially no uncertainty on their value. Note that in any case the result is perfectly consistent with HR.

- Q10: *The discussion deserves more rigor and more biological examples to demonstrate the usefulness of the approach (apart from lowering the computation time for similar approximation of marginals). What do the authors mean by sufficient statistics of a Gaussian distribution? In what sense are the $N \times N$ and $M \times M$ matrices equivalent?*

A10: First, we underline that lowering of computing time for very large metabolic networks, like the *RECON1* model, is an important point in itself, as the computing time needed for the uniform sampling of HR is close to being unfeasible: it required for us about 20 days to obtain the best estimates with $T = 2.6 \times 10^9$ sample points, and the correlations seems to be still on an increasing trend.

Concerning the biological examples, we have decided to follow the referee’s suggestion and develop in detail how to incorporate the information about an experimentally measure of the distribution of the biomass flux (as Reviewer 1 suggested in Q8 of the section “Comments to the method”). The update equations appearing in Appendix G are now reported in the “Methods” section where, moreover, we have better explained how to derive them. To give a biological example, we have applied this slightly modified algorithm to the *iJR904* model for *Escherichia Coli* in some specific conditions. The complete discussion is now reported in the new Subsection “Study of *Escherichia Coli* metabolism for a constrained growth rate” of the “Results” section.

As the reviewer points out, the expression “sufficient statistics” is not clear and unneeded. We have replaced it by “the optimization of two parameters, the mean and the variance of a Gaussian distribution”.

Regarding the “ $N \times N$ and $M \times M$ matrices”, we think that the expression of the original work can be misunderstood. With “equivalent” we meant that the $N \times N$ matrix Σ can be written, using Woodbury formula, in the following way:

$$\Sigma = (\beta \mathbf{S}^t \mathbf{S} + \mathbf{D})^{-1} \quad (1)$$

$$= \mathbf{D}^{-1} - \beta \mathbf{D}^{-1} \mathbf{S}^T (\mathbf{1}_{M \times M} + \beta \mathbf{S} \mathbf{D}^{-1} \mathbf{S}^T)^{-1} \mathbf{S} \mathbf{D}^{-1} \quad (2)$$

where $\mathbf{1}_{M \times M}$ is the $M \times M$ identity matrix. Thus, instead of inverting the original $\beta \mathbf{S}^t \mathbf{S} + \mathbf{D}$, we can invert $\mathbf{1}_{M \times M} + \beta \mathbf{S} \mathbf{D}^{-1} \mathbf{S}^T$ that has dimension $M \times M$. Although we believe that this equality can be very helpful, the statement in the Discussion part and the relative paragraph in Appendix E have been omitted since all results presented in this paper are obtained through the inversion of the original matrix.

- Q11: *Parts of section IV.B cannot be understood without reading the appendices as clarification of the notation used.*

A11: We agree with the referee; the paragraph is not clear enough. We have postponed the discussion about the EP free energy functional and its link to other inference approaches in Appendix B where this functional is derived in detail.

Comments to the Method

- Q1: *Why is the normalization constant from Eq. II.2 neglected if the limits of II.5 in limits of β to infinity?*

A1: In the original submission we omitted terms that do not depend on ν (and thus were not needed in the final expression). For the sake of clarity, we corrected Eq. (II.2) to include the normalization explicitly:

$$P(\nu|\mathbf{b}) = \frac{1}{P(\mathbf{b})} \left(\frac{\beta}{2\pi} \right)^{\frac{M}{2}} e^{-\frac{\beta}{2}(\mathbf{S}\nu - \mathbf{b})^T(\mathbf{S}\nu - \mathbf{b})} \prod_{n=1}^N \psi_n(\nu_n) \quad (3)$$

In addition, to underline the dependence on the unknowns variables, we have introduced the sentence “Neglecting all the terms that do not depend on ν , the posterior takes the form of $P(\nu|\mathbf{b}) \propto e^{-\frac{\beta}{2}(\mathbf{S}\nu - \mathbf{b})^T(\mathbf{S}\nu - \mathbf{b})} \prod_{n=1}^N \psi_n(\nu_n)$ where we have not explicitly reported the normalization constant”. Note that the normalization constant hidden in the proportionality symbol does indeed depend on β . Note also that we do not perform the $\beta \rightarrow \infty$ limit explicitly, but just employ a very large value of β .

- Q2: *It seems that the fluxes shown in Figure II.1 involve exchange reactions. How does the first local approach behave for internal reactions?*

A2: The results remain qualitatively the same. We report here, in Figure 1, the results of EP against the non-adaptive approach (*first local approach* in the old version of the manuscript) for a subset of internal/exchange reactions (these have been chosen randomly, the behavior is similar in other cases. We find that the non-adaptive approach generally overestimates the variances, and is very inaccurate for the means)

- Q3: *What are the rules for numerical convergence used in this study (lines 151- 152).*

A3: To complete the discussion we have added a reference to the subsection “Numerical results” where the criterion is described. The following sentence has been added to line 152: “Further technical details about the convergence are reported in Subsection II.C 1”.

- Q4: *Doesn't Figure II.1 (b) report the accuracy rather than the efficiency of the approach?*

A4: Indeed. We replaced in line 159 “Efficiency” by “accuracy”.

- Q5: *Most critically, how does one chose the value of beta a priori? Is there any guideline for this important issue? If not resolved, the approach will have the same issues as the criticized HR whose convergence depends on the scenarios investigated (as stated in the introduction).*

A5: For the results reported in the work we have set $\beta = 10^{10}$ except for the biggest model, RECON1, where $\beta = 10^9$. This has been done because EP did not converge for $\beta = 10^{10}$ on the RECON1 model. We show in Figure 2 that results do not significantly change for $\beta \in [10^7, 10^{10}]$, so our guideline is to always use a very large value for this parameter, for instance $\beta = 10^{10}$ and, in case of non-convergence, decrease it. We would like to stress that, even though convergence is not guaranteed, the tuning of this parameter only requires few runs

Figure 1. Marginals for internal/exchange reactions of ecoli-core model for Escherichia Coli. (a) Non-adaptive approach (b) Expectation Propagation

of EP. The total time spent for the setting of β is thus of the order of minutes (or seconds in most cases) that is negligible with respect to the convergence time of a typical HR run investigated in this work. We added these considerations to the manuscript in the section “Numerical results for large scale metabolic networks”.

- Q6: *The error in the means and the variances should be a function of \mathbf{a} and \mathbf{d} , the parameters to be estimated (see lines 178).*

A6: In the first submission we have erroneously used \mathbf{b} instead of \mathbf{d} , which we have now corrected. Actually, a more consistent way of estimating the error (which we had actually used in the simulations) is to compute differences on the first and second moment of the (truncated) marginal distribution on successive time-steps, as these are the quantities we use for the comparison. We have rephrased this part.

- Q7: *The discussion is convoluted when it comes to the treatment of the ranges for b allowed! Why is a range from 10^{-50} to 10^{50} considered and how is this precision achieved in the implementation? It is clear that a simple variant of Flux Variability Analysis can determine the range of values that a flux is allowed to take in the feasible space. Why not use these as bounds?*

A7: This was an unfortunate typo. We have erroneously indicated the variances of the approximation with \mathbf{b} instead of \mathbf{d} as for Q6. All the discussion refers to bounds on the parameters of the approximation and does not involve the intakes/uptakes \mathbf{b} nor the bounds on the fluxes that are provided by the model and can take any real number.

- Q8: *The authors should provide one biologically motivated example where experimental evidence is encoded (e.g. confidence intervals of flux estimates from labeling studies in different model species).*

A8: We have discussed in detail how to enforce an experimental profile in our algorithm in section “Study of Escherichia Coli metabolism for a constrained growth rate”.

- Q9: *The definition of \mathbf{D} does not match the standard definition of a diagonal matrix!*

A9: \mathbf{D} is indeed a diagonal matrix from which we had only specified the diagonal terms. We have completed the description of the matrix \mathbf{D} by specifying that non-diagonal terms are 0.

Figure 2. Left: Pearson coefficients computed between averages and variances for $\beta = 10^{10}$ against results obtained for $\beta = [10^7, 10^8, 10^9]$. Right: scatter plots of the averages (up) and variances (down) computed for $\beta = 10^{10}$ (y-axis) against $\beta = [10^7, 10^8, 10^9]$ (x-axis)

- Q10: *The S used for stoichiometric matrix in the introduction is different from that in Eq. IV.3. Please, unify.*
A10: To unify the notation we have decided to use the symbol **S** for the stoichiometric matrix.
- Q11: *Are there any fluxes for which convergence is achieved easier than for others?*
A11: Indeed some fluxes seem to reach convergence after few iterations of the algorithm. To give an example, we report a plot of the number of converged fluxes as a function of the iterations for the iJR904 model. The first three fluxes that converged are EX_met_D(e), METDabc, and DOGULNR while the last three are EX_pi(e), PIt2r, and EX_h2o(e).
- Q12: *It is desirable to include a detailed discussion about the assumption $d_n \neq 0$ in line 366 (to derive Eqs. A.11*

Figure 3. Number of converged fluxes against iterations for iJR904

and A.12), especially in light of including additional constraints (which render it more likely that this condition is violated).

A12: Violation of the condition $d_n > 0$ can indeed occur during the iteration, and it is a sign that the EP approximation cannot capture the true distribution (maybe because it is too constrained as the referee suggests). For example, one simple case in which this surely occurs is when $\nu_n^{inf} = \nu_n^{sup}$ (but thi case of course can be eliminated by simplifying the system). In practice, the truncation bounds $[10^{-50}, 10^{50}]$ we apply on d_n ensure that this singularity is never propagated numerically, but of course the truncation may indeed be source of inaccuracy. What we observe is that at convergence of the algorithm, none of the d_n takes the boundary value 10^{-50} : these boundaries are only reached during the iterations.

RESPONSE TO REVIEWER 2

- Q1: *The technique of Flux Balance Analysis should be mentioned explicitly in the abstract (not just as linear programming).*

A1: We have added “(as Flux Balance Analysis)” in line 20

- Q2: *Line 25: Spell out TAP*

A2: We eliminated TAP as it was not essential.

- Q3: *Line 27: Cavity \rightarrow cavity*

A3: “Cavity” changed in “cavity”;

- Q4: *Line 47: $\nu,^{inf}\nu^{sup} \rightarrow \nu^{inf}, \nu^{sup}$*

A4: Comma shifted: ν^{inf}, ν^{sup} ;

- Q5: *Line 49: After the sentence “Empirically, it turns out that $N < M$ so that the system is typically severely underdetermined”, it should be perhaps appropriate to give some examples of values N and M for some metabolic networks, to give an idea to the reader.*

A5: Agreed. We changed to “Empirically, it turns out that $N > M$ so that the system is typically severely under-determined. For instance, for the RECON1 model of Homo Sapiens $N = 2469$ and $M = 1587$ ” N and M are typically of the same order of magnitude, so we have changed \gg in $>$.

- Q6: *Line 52-53. I wonder whether the author agree with me in that the following reference could be useful for some readers regarding Flux Balance Analysis:*

* *Kauffman, Kenneth J., Purusharth Prakash, and Jeremy S. Edwards. “Advances in flux balance analysis.” Current opinion in biotechnology 14.5 (2003): 491-496.*

A6: We agree, and added the reference to the bibliography;

- Q7: Line 64: "Basically consists" → "Basically, it consists"

A7: Done.

- Q8: Line 79: The following references seem to be missing (importantly, the second one):

- * Font-Clos, Francesc, Francesco Alessandro Massucci, and Isaac Pérez Castillo. "A weighted belief propagation algorithm for estimating volume-related properties of random polytopes." *Journal of Statistical Mechanics: Theory and Experiment* 2012.11 (2012): P11003.
- * Fernandez-de-Cossio-Diaz, Jorge, and Roberto Mulet. "Fast inference of ill-posed problems within a convex space." *arXiv preprint arXiv:1602.08412* (2016).

A8: We thank the referee, as both are indeed important references; we added them to the bibliography.

Formulation of the problem

- Q9: In the section "Formulation of the problem" (from line 102), I have the following suggestions/comments:

- I think that the authors should somewhat clarify early how the bounds on the fluxes is taken into account. I am worried that a reader, might start wondering about this from line 105 forward. Perhaps, the following rephrase on line 104 might be useful: "As standard, and ignoring for the moment the bounds in the fluxes, we can define a quadratic energy function...", or something similar.

A9:

- We agree with the reviewer and we modified the phrase to be more explicit as follows: "For a vector of fluxes satisfying bounds [L.2], we can define a quadratic energy function $E(\boldsymbol{\nu})$ whose minimum(s)..."

First local approach

- Q10: In the section "First local approach" (from line 121), I have the following suggestions/comments:

- Although it is implicit in the form of the Gaussian prior (line 123), it should be made explicit that, with this choice, the fluxes are now unbounded.
- In Eq. (II.8) the symbol $\forall n$ is used, while in line 114 the symbol $n \in \{1, \dots, N\}$ is used. Please, unify presentation of the notation.

A10:

- We underlined this "feature" of the approximation by adding the sentence: "Notice that in this approximation fluxes result unbounded"
- We have unified the notation, using $n \in \{1, \dots, N\}$ instead of $\forall n$

Expectation Propagation

- Q11: In the section "Expectation Propagation" (from line 131), I have the following suggestions/comments:

- Line 132: "Expectation Propagation [19]" → "Expectation Propagation (EP) [19]"
- Eq (II.9) is not clear. A better alternative is:

$$Q^{(n)}(\boldsymbol{\nu}|\mathbf{b}) \doteq \frac{1}{Z_{Q^{(n)}}} e^{-\frac{\beta}{2}(\mathbf{S}\boldsymbol{\nu}-\mathbf{b})^T(\mathbf{S}\boldsymbol{\nu}-\mathbf{b})} \psi_n(\nu_n) \prod_{m \neq n} \phi_m(\nu_m) \quad (4)$$

- Moreover the sign \doteq has various meanings in Mathematics, and for some readers it may not be clear from the context.
- In reference [24] of line 143, it is said: "Of course, it is an extensive number of "intractable" priors that makes the computation intractable, not a single one!". Does this mean that a possible generalization of the EP method would be to include an intensive number of intractable priors, viz

$$Q^{(n_1, \dots, n_L)}(\boldsymbol{\nu}|\mathbf{b}) \doteq \frac{1}{Z_{Q^{(n)}}} e^{-\frac{\beta}{2}(\mathbf{S}\boldsymbol{\nu}-\mathbf{b})^T(\mathbf{S}\boldsymbol{\nu}-\mathbf{b})} \psi_{n_1}(\nu_{n_1}) \dots \psi_{n_L}(\nu_{n_L}) \prod_{m \neq (n_1, \dots, n_L)} \phi_m(\nu_m) \quad ? \quad (5)$$

According to your expertise, how will this affect the viability of your method for moderate values of L ? I guess that it must be important which set of fluxes (n_1, \dots, n_L) are chosen in this generalization (ie. highly correlated fluxes)

- (e) The panels (a) and (b) of figure II.1 should be made bigger (space permitting) and the labels in the axes also bigger, so it improves readability.
- (f) If we take e.g. $L = 2$ in Eq. (2) of this report and take appropriate choice of pairs of fluxes, could we further improve the results of panel (b) in figure II.1.
- (g) On panel (b) of figure II.1 the authors compare the results of EP with HR. However, I believe they should also compare with what it is the state-of-the-art of BP algorithms presently, which is the one presented in
- * Fernandez-de-Cossio-Diaz, Jorge, and Roberto Mulet. "Fast inference of ill-posed problems within a convex space." *arXiv preprint arXiv:1602.08412* (2016).

A11: We have accepted all the suggestions in (a), (b) and (e).

- (c) We simply replaced the symbol " \doteq " with " $=$ " as it is clear from the sentence that it is a definition.
- (d) Of course this generalization can lead to interest results and it is something that must be taken into account for future developments. We want to underline some points:
 - Suppose $L = 2$. The computation of the moments of the tilted distribution with two intractable priors becomes slightly harder. Mathematically speaking, for any pair (n, m) of fluxes, we need to compute the expectations $\langle \nu_n^\alpha \rangle_{Q^{(n,m)}}$, $\langle \nu_m^\alpha \rangle_{Q^{(n,m)}}$ for $\alpha = \{1, 2\}$ over the two-fluxes tilted distribution

$$Q^{(n,m)}(\boldsymbol{\nu}|\mathbf{b}) \propto e^{-\frac{\beta}{2}(\mathbf{b}-\mathbf{S}\boldsymbol{\nu})^T(\mathbf{b}-\mathbf{S}\boldsymbol{\nu})} \psi_n(\nu_n) \psi_m(\nu_m) \prod_{l \neq \{n,m\}} \phi_l(\nu_l) \quad (6)$$

Consider, for instance, the computation of the first moment $\langle \nu_n \rangle_{Q^{(n,m)}}$

$$\langle \nu_n \rangle_{Q^{(n,m)}} \propto \int_{\nu_n^{min}}^{\nu_n^{max}} d\nu_n \nu_n \int_{\nu_m^{min}}^{\nu_m^{max}} d\nu_m e^{-\frac{1}{2}(\boldsymbol{\nu}^{(nm)} - \bar{\boldsymbol{\nu}}^{(nm)})^T \boldsymbol{\Sigma}_{(nm)}^{-1} (\boldsymbol{\nu}^{(nm)} - \bar{\boldsymbol{\nu}}^{(nm)})} \quad (7)$$

where we have used the two components vectors $\boldsymbol{\nu}^{(nm)} = (\nu_n, \nu_m)$, $\bar{\boldsymbol{\nu}}^{(nm)} = (\bar{\nu}_n, \bar{\nu}_m)$ and a 2×2 matrix $\boldsymbol{\Sigma}_{(nm)}^{-1} = \begin{pmatrix} \Sigma_{nn} & \Sigma_{nm} \\ \Sigma_{mn} & \Sigma_{mm} \end{pmatrix}$. If we perform the integration in (7) with respect to ν_m we obtain something which depend on the erf $\left(\frac{\nu_n - \bar{\nu}_n}{\Sigma_{nn}}\right)$. The integral is now the computation of the first moment of a distribution which contains the error function. To the best our knowledge there is no closed expression to exactly perform this integration. We could numerically compute it (for instance via Monte-Carlo methods) with the drawback of severely slowing down the algorithm, or perform some expansion.

- Notice that when we perform the matching of the moments of the distribution in (6) and $Q(\boldsymbol{\nu}|\mathbf{b})$ we have, in principle, 5 equations for 4 unknowns. The fifth missing equation regards the expectation $\langle \nu_n \nu_m \rangle_{Q^{(n,m)}}$ with which we do not have any associated parameter. To take into account this information, we could generalize even more our approximation, and couple the N fluxes using some number $k \leq \binom{N}{2}$ of bivariate Gaussians or eventually using multivariate Gaussians. Unfortunately, it is not clear how to efficiently manage this approximation but we do not exclude to develop more this point in future works. We just mention that something has been already done in this direction in

* Qi, Yuan and Minka, T. P. "Tree-structured approximations by expectation propagation". *Advances in Neural Information Processing Systems (NIPS)*

Numerical results

- Q12: In the subsection "Numerical results" (from line 163)

- (a) *The order in the discussion between lines 168-172 is a bit confusing. Perhaps it would be more appropriate to discuss the part regarding the preprocessing the network first and then discuss how HR and EP are run in the reduced networks.*
- (b) *In line 182: with MS, do you mean EP?*
- (c) *In line 193: "Person's" → "Pearson's"*
- (d) *In line 190-191: "This coefficient is equals..." → "This coefficient is equal..."*
- (e) *In line 195: "Per each" → "For each"*
- (f) *In line 202: "Results suggest that the more T grows the more the points, either the means or the variances, are correlated" → "Results suggest that as T grows, more points, either the means or the variances, become correlated"*

A12: (c), (e), (f) Thank you for the corrections;

- (a) According to the suggestion, we have shifted lines 168-170 after the explanation of pre-processing techniques.
- (b) Yes, we do mean EP (fixed);
- (d) We erased the entire phrase since the Pearson's coefficient is positive (negative) for positive (negative) correlations; it is not exactly equal to +1 (-1).

Discussion

- Q13: *In the section "Discussions" (from line 209)*

- (a) *In line 209 "Discussions" → "Discussion".*
- (b) *In line 236, the power EP algorithm is mentioned for the first time. Is this a new implementation? Is this a known method? If the latter, can a reference be provided?. This paragraph does not seem to add anything substantial to the paper. If the authors agree, perhaps it should be removed altogether from the paper (and the corresponding the appendix G). Otherwise, please give a some argument about the importance of such a variation.*

A13:

- (a) We have changed the title of the section.
 - (b) Power EP consists in slightly modified updating of the parameters \mathbf{a} and \mathbf{b} ; the original work is presented in:
- * *Tom Minka and others. "Divergence measures and message passing." Technical report, Technical report, Microsoft Research, 2005.*

Basically, instead of minimizing, at each step, the Kullback-Leibler divergence between the full approximate distribution and the tilted one, one has to minimize a general α -divergence that is formally defined, for two distributions $p(x)$, $q(x)$, $x \in X$ as

$$D_{\alpha}(p||q) \equiv \frac{\int_X \alpha p(x) + (1 - \alpha) q(x) - p(x)^{\alpha} q(x)^{1-\alpha} dx}{\alpha(1 - \alpha)} \quad (8)$$

In the limit $\alpha \rightarrow 1$ one recovers the "inclusive" KL distance and so EP. This method sometimes provides better results then the standard EP algorithm, but it was not the case for this model. As the reviewer suggested, it is better to omit this part and Appendix G (of the old manuscript) and so we did.

Method

- Q14: *In the section "Methods" (from line 260)*

- (a) *I am not sure the concept of exclusive and inclusive KL divergence is known. Please make sure this is explained in the references provided or explain otherwise.*

A14: The definition of these two quantities is already on the text. Nevertheless, we have added the following line and reference: "A complete discussion about "exclusive" and "inclusive" divergences are provided in [*]".

* *Tom Minka and others. Divergence measures and message passing. Technical report, Technical report, Microsoft Research, 2005.*

Finally, we remain at your entire disposal for any further clarification you might need.

Best regards,

Alfredo Braunstein, Anna Paola Muntoni, Andrea Pagnani

Reviewers' comments:

Reviewer #1 (Remarks to the Author):

Comments to Braunstein et al. „An analytic approximation of the feasible space of metabolic networks“

The authors have provided substantial changes to the manuscript by: (1) rewriting major portions of the text to make few sections more accessible to the readership of Nature Communications, (2) showing the benefit of the approach over the existing alternatives, particularly in the case when additional constraints from experimental measurements are available, and (3) re-iterating the benefits from the first version by considering more realistic parameter values.

Nevertheless, in implementing the changes, still a major chunk of the manuscript, particularly the introduction and some sections of the results remain very difficult to follow. This may in part be due to the issues with wording, which render some of the claims confusion or incorrect. Therefore, while I find that the manuscript is technically sound, the presentation will have to be streamlined to correct some issues which I identify below within each of the respective sections.

Abstract

The abstract still makes it very difficult to delineate what is the main contribution of the approach. For instance, there is no need to define metabolic fluxes in the opening statement. What does it mean to characterize the space of fluxes (wouldn't this be done by finding all the generators? This is just one possible way to understand that statement). For a general readership, it will not be clear what is sampled and why the sampling is done, as is the introduction of the cavity method. Instead of comparisons to the cavity method, the statement "Here we present ..." could be further strengthened. Most critically, I do not see why the measured flux distributions are "efficiently measured" – the existing methods for experimental flux profiling are not efficient as they require a lot of data gathering followed by model fitting (so, we talk about flux estimates rather than measurements, since they come from a model!) The statement about the extensive comparative analysis is not particularly illuminating, as it does not specify the key advantages.

Introduction

1. Nutrients are also used for replication, defense, and other cellular tasks. Please, rephrase line 28.
2. The mention of reaction constants already assumes that you talk about reaction rates modeled as functions of enzyme and metabolite concentrations, by say, mass action kinetic. This is not obvious to a general reader.
3. What do you mean by stating "assume a steady-state regime in the system where fluctuation of metabolite concentration remain constant over time"? In a steady state, the derivative of concentration change is zero, so no fluctuations occur.
4. By coefficients of the corresponding metabolite in all reactions, on line 37, the authors mean "stoichiometric coefficient". Please, add the missing word.
5. what do you mean by "severely underdetermined"? A system is or it is not underdetermined.
6. FBA aims a predicting growth, corresponding to the rate of the biomass reaction; hence, this linear program has one solution for the objective function of maximizing biomass rate; however, there could be several flux distributions which amount to the same optimal value for the objective, resulting in a space of alternative optimal flux distributions. The goal of FBA

has never been “to characterize” the space of alternative optimal flux distributions, so rephrasing of the paragraph on lines 48 – 54 is needed.

7. What do the mixing times refer to on line 65?

8. What does it mean for an assumption to be uncontrolled on line 78?

9. change “strong correlated stoichiometric matrices” to “strongly row-correlated stoichiometric matrices”

10. The sentences about the discussion section on line 96 – 98 are not very informative and can be removed.

11. The opening paragraph on pp. 5 is very difficult to follow, largely due to the first sentence.

12. The section on Numerical results should be rewritten – the paragraph on lines 172 – 176 is a standard in detecting and removing all blocked reactions! So, referring to blocked reactions and providing a classical reference would suffice. The method would also suffer from fluxes which are constant irrespective of the point in the subspace of the convex cone explored. Did the authors ensure that these are not present in the current models?

13. ϵ in the equation between lines 189 and 190 must depend on t !

14. The equation II.2 is standard and can be removed.

15. What is meant by “fairly sample” the polytope on line 204?

16. I find Figure II.2 still too large; reconsider some other presentation of these important findings, perhaps for fewer values of T .

17. Figure II.3 is not very informative; maybe mark the point corresponding to biomass of 2 h^{-1} . By the way, the units for the biomass flux are wrong, as it usually comes in $mol (gDW)^{-1} h^{-1}$!

18. The paragraph on lines 236 – 247 is too technical and can be considerably simplified by a brief description of what is shown in Fig. II. 4.

19. The paper will benefit from showcasing the results on lines 249 – 286, which are now not too prominent.

Discussion

20. It should avoid pointers to Appendices and should be as self-contained as possible, while highlighting the advantages of the approach.

Methods

21. I cannot follow the definition of D !

Appendices H and I should be Supplementary results (if such a section is allowed).

Reviewer #2 (Remarks to the Author):

I am pleased with the changes made. The article is ready for publication.

Dear Editor,

We thank both reviewers for their critical reading of the revised version of the manuscript. We addressed the issues raised by Referee 1 to the best of our abilities, and indicate the changes and response to comments below.

- *Q: Abstract. The abstract still makes it very difficult to delineate what is the main contribution of the approach. For instance, there is no need to define metabolic fluxes in the opening statement. What does it mean to characterize the space of fluxes (wouldn't this be done by finding all the generators? This is just one possible way to understand that statement). For a general readership, it will not be clear what is sampled and why the sampling is done, as is the introduction of the cavity method. Instead of comparisons to the cavity method, the statement "Here we present ..." could be further strengthened. Most critically, I do not see why the measured flux distributions are "efficiently measured" – the existing methods for experimental flux profiling are not efficient as they require a lot of data gathering followed by model fitting (so, we talk about flux estimates rather than measurements, since they come from a model!) The statement about the extensive comparative analysis is not particularly illuminating, as it does not specify the key advantages.*

A: We completely rewrote the abstract with the referee suggestions in mind. While we agree that “characterizing the space of fluxes” was not well defined, it could be argued that some previous knowledge of the reader needs to be assumed (e.g. the referee rightly suggests to remove the definition of fluxes). So in this fine line in which we need to make some compromises we proceeded as follows: **(a)** As suggested, we removed the definition of fluxes. **(b)** We clarified the meaning of the *characterization* of the space by explaining the use of individual tools: FBA can be used to find extreme values of each flux and HR to find flux marginals; this helps both to give examples of what we mean with “characterizing” and for distinguishing between the scope of different standard tools (clarifying also the use of sampling). **(c)** We made more precise the statement “Here we present...” by clarifying that the method is used to compute flux marginals. **(d)** We changed “fix efficiently measured distributions of observables” to “efficiently fix empirically estimated distribution of fluxes” which was what we really meant, incorporating also another suggestion of the referee. Note that Appendix F (old Appendix G) is essentially devoted to justify the word “efficiently”, as this constrained computation is hard or even impossible to achieve with an a-posteriori weighting of HR samples.

1. *Q: Introduction: Nutrients are also used for replication, defense, and other cellular tasks. Please, rephrase line 28.*
A: We rephrased the sentence as suggested.
2. *Q: The mention of reaction constants already assumes that you talk about reaction rates modeled as functions of enzyme and metabolite concentrations, by say, mass action kinetic. This is not obvious to a general reader.*
A: Indeed. We removed the sentence altogether, as it was not essential and possibly confusing.
3. *Q: What do you mean by stating “assume a steady-state regime in the system where fluctuation of metabolite concentration remain constant over time”? In a steady state, the derivative of concentration change is zero, so no fluctuations occur.*
A: We removed “fluctuation of”
4. *Q: By coefficients of the corresponding metabolite in all reactions, on line 37, the authors mean “stoichiometric coefficient”. Please, add the missing word.*
A: We added the missing word.
5. *Q: What do you mean by “severely underdetermined”? A system is or it is not underdetermined.*
A: We intended to mean that the matrix rank is *much* smaller than the number of variables, so the dimension of the solution space is typically large. To avoid confusion, we removed the word “severely” as this is clear from the example that follows.
6. *Q: FBA aims a predicting growth, corresponding to the rate of the biomass reaction; hence, this linear program has one solution for the objective function of maximizing biomass rate; however, there could be several flux distributions which amount to the same optimal value for the objective, resulting in a space of alternative optimal flux distributions. The goal of FBA has never been “to characterize” the space of alternative optimal flux distributions, so rephrasing of the paragraph on lines 48 – 54 is needed.*
A: We removed the part which might have alluded to the limits of FBA into: “However, if one is interested in describing

more general growth conditions, or is interested in other fluxes than the biomass, different computational strategies must be envisaged”.

7. Q: *What do the mixing times refer to on line 65? .*

A: The *mixing time* is a concept in Markov Chain Theory which basically measures the number of elementary chain steps required to approximate the stationary probability distribution induced by the Markov process. We rephrased line 65 into the more accessible : “[...] often establishing in practice how long a simulation should be run and how frequently the measurement should be taken [...]”

8. Q: *What does it mean for an assumption to be uncontrolled on line 78?*

A: We changed the sentence into: “Unfortunately, neither assumption is really fulfilled by large-scale metabolic reconstructions.”

9. Q: *Change “strong correlated stoichiometric matrices” to “strongly row-correlated stoichiometric matrices”*

A: Changed.

10. Q: *The sentences about the discussion section on line 96 – 98 are not very informative and can be removed.*

A: We removed the sentences.

11. Q: *The opening paragraph on pp. 5 is very difficult to follow, largely due to the first sentence.*

A: We simplified the starting sentence to “Expectation Propagation (EP) [26] is an efficient technique to approximate intractable (i.e. impossible to compute analytically) posterior probabilities.” and removed some superfluous historical details.

12. Q: *The section on Numerical results should be rewritten – the paragraph on lines 172 – 176 is a standard in detecting and removing all blocked reactions! So, referring to blocked reactions and providing a classical reference would suffice. The method would also suffer from fluxes which are constant irrespective of the point in the subspace of the convex cone explored. Did the authors ensure that these are not present in the current models?*

A: We removed the sentence altogether making a reference to: Henry, Christopher S., et al. "High-throughput generation, optimization and analysis of genome-scale metabolic models." Nature biotechnology 28.9 (2010): 977-982. Regarding fluxes that can only take one value (as ATP maintenance or blocked ones), note that the algorithm has actually no particular problem dealing with them. The reduction is done just for the sake of efficiency, as it is very fast and reduces the problem size.

13. Q: *ϵ in the equation between lines 189 and 190 must depend on t !*

Correct, we added the dependency on t .

14. Q: *The equation II.2 is standard and can be removed.*

A: We think that Reviewer 1 is pointing at Eq. II.12 (Pearson coefficients). We removed the equation.

15. *What is meant by “fairly sample” the polytope on line 204?*

We entirely changed the sentence, focusing on the evidence, without referring to the characteristics of the sampling.

16. Q: *I find Figure II.2 still too large; reconsider some other presentation of these important findings, perhaps for fewer values of T .*

A: Following the suggestion, we kept only four values of T .

17. Q: *Figure II.3 is not very informative; maybe mark the point corresponding to biomass of 2 h⁻¹. By the way, the units for the biomass flux are wrong, as it usually comes in mol (gDW)⁻¹ h⁻¹!*

A: We removed the figure as we needed only one point as the referee points out. With respect to the units, the Referee is correct, we were referring to the growth rate and not to the biomass flux. We corrected this and another unit mistake.

18. Q: *The paragraph on lines 236 – 247 is too technical and can be considerably simplified by a brief description of what is shown in Fig. II. 4.*

A: We simplified the introductory description, but we left it because in our opinion it is needed to understand the results in the Figure.

19. *Q: The paper will benefit from showcasing the results on lines 249 – 286, which are now not too prominent.*

A: We pointed to these results at the end of the Introduction.

20. *Q: Discussion. It should avoid pointers to Appendices and should be as self-contained as possible, while highlighting the advantages of the approach.*

A: We removed pointers to the Appendices in the Discussion. We removed also Appendix B “Expectation propagation free energy functional” which was only referred to on the Discussion, but was not really necessary.

21. *Q: Methods. I cannot follow the definition of D ! Appendices H and I should be Supplementary results (if such a section is allowed).*

A: \mathbf{D} is the following matrix:

$$\mathbf{D} = \begin{pmatrix} \frac{1}{d_1} & 0 & \dots & & \dots & 0 \\ 0 & \ddots & \ddots & & & \vdots \\ \vdots & \ddots & \frac{1}{d_{n-1}} & & & \\ & & & 0 & & \\ & & & & \frac{1}{d_{n+1}} & \ddots & \vdots \\ \vdots & & & & \ddots & \ddots & 0 \\ 0 & \dots & & \dots & 0 & \frac{1}{d_N} \end{pmatrix}$$

Possibly the confusion is due to the fact that we omitted the explicit dependence of the matrix \mathbf{D} on the variable index n to simplify the notation. We added a sentence in the manuscript to clarify this aspect. We moved Appendices H and I (Now G and H) to an independent Supplementary Results document.

REVIEWERS' COMMENTS:

Reviewer #1 (Remarks to the Author):

The authors have addressed all my final comments appropriately. The abstract reads well and is accessible for a general audience.